# CLASS-INCREMENTAL LEARNING USING GENERATIVE EXPERIENCE REPLAY BASED ON TIME-AWARE REGULARIZATION

## ABSTRACT

Learning new tasks accumulatively without forgetting remains a critical challenge in continual learning. Generative experience replay addresses this challenge by synthesizing pseudo-data points for past learned tasks and later replaying them for concurrent training along with the new tasks' data. Generative replay is the best strategy for continual learning under a strict class-incremental setting when certain constraints need to be met: (i) constant model size, (ii) no pre-training dataset, and (iii) no memory buffer for storing past tasks data. Inspired by the biological nervous system mechanisms, we introduce a time-aware regularization method to dynamically fine-tune the three training objective terms used for generative replay: supervised learning, latent regularization, and data reconstruction. Experimental results on major benchmarks indicate that our method pushes the limit of a brain-inspired continual learner under such strict settings, improves memory retention, and increases the average performance over continually arriving tasks.

## 1 INTRODUCTION

Incremental learning is a continual learning setting, where new novel classes are encountered over time Van de Ven & Tolias (2019). The goal in class incremental learning is to enable a base model to learn new classes that arrive sequentially. Catastrophic forgetting (French, 1999) is a major challenge in this setting because when the model is updated to learn new classes, its performance on the past learn classes would degrade. Experience replay (Schaul et al., 2015) is a major approach to addressing catastrophic forgetting. The idea is to stor and then replay representative samples of past classes along with the samples of new classes to enforce the model to maintain its performance in past classes. When storing data is not feasible, e.g., due to data privacy, generative replay is applicable (Shin et al., 2017). The idea is to enable the model to generate pseudo-data points that resemble the original data for learned classes. Existing generative replay approaches with superior performance (Van de Ven et al., 2020) are brain-inspired, where modules and mechanisms similar to the interactions between the prefrontal cortex (PFC) and hippocampus are designed to mimic the short-term cognitive function and long-term memory to mitigate catastrophic forgetting.

Despite being effective, the existing methods often use a simple static weighting mechanism between these two brain components, ignoring the fact that the bidirectional information pathways between them are changing through time. In contrast, the brain is a dynamic system, with research showcasing the importance of feedback connections in information processing and memory consolidation (Thierry et al., 2000). Specifically, there is a bi-directional interplay between the PFC and the hippocampus, where the PFC not only gathers information from the hippocampus but also returns feedback which can modulate the functionality of the hippocampus (Eichenbaum, 2000; Preston & Eichenbaum, 2013). For instance, the synaptic plasticity of the hippocampus, referring to its adaptive nature in response to experiences, plays a pivotal role in memory consolidation. Under certain circumstances, such as during attentive learning or active memory recall, feedback from the PFC to the hippocampus can lead to enhanced synaptic plasticity. This in turn increases the likelihood of short-term memory transitions into long-term memory (Wang et al., 2010). Although methods like brain-inspired replay (Van de Ven et al., 2020) implement the bidirectional feedback pathways through a shared network between these two components, the above-mentioned biological changes under different circumstances are not reflected in the objective design. We thus deem it important to

also adjust the objectives in addition to architecture change. This adjustment is also brain-inspired and proves to be highly effective in generative replay in our later experiments.

Moreover, such dynamic feedback control on objectives is crucial from a model optimization perspective. During generative replay training, replayed data from recent or distant tasks are optimized equally with real new data in the discriminator. The training objective to distinguish these classes will inevitably consider the time-related features such as over-regularization and distance from the real data distribution, which accumulated during the sample-optimize-sample loop through time. Passing this time-related information to the generative model and modifying the confidence level of each sample will improve memory consolidation. More specifically, the closer to the real data distribution, the more confidence we should be able to consolidate into the memory. Bearing these in mind, we look into how we can use the inherent time-related information from the discriminator to dynamically adjust the objective of the generator. Our specific contributions include:

- We design an objective scheduling mechanism that depends on an inferred time, which we claim to be implicit in the discriminator's prediction.

- We improve the performance of the existing SOTA brain-inspired continual learning methods by a substantive margin without adding time, or space complexity.

- We improve the existing brain-inspired replay methods' memory retention ability by providing more diverse replayed generated samples of previously seen classes.

- We demonstrate that our method is closely related to the activity in the brain qualitatively.

## 2 RELATED WORK

There has been many works on class incremental learning setting. Recent works in this area can be grouped into five classes based on the used strategies to address catastrophic forgetting: (i) data replay (Bang et al., 2022; 2021; Mai et al., 2021; De Lange & Tuytelaars, 2021; Van de Ven et al., 2020; Rolnick et al., 2019; Chaudhry et al., 2018; Shin et al., 2017) which involves storing a subset of training data and then replaying them back, (ii) growing network (Zhou et al., 2023; Douillard et al., 2022; Wang et al., 2022; Yan et al., 2021) which involves learning the new classes through new added network weights, (iii) model regularization (Yang et al., 2021; Aljundi et al., 2019; Yang et al., 2019; Lee et al., 2019a; 2017; Aljundi et al., 2018; Kirkpatrick et al., 2017; Zenke et al., 2017) which involves identifying important network weights and then consolidating them when learning new classes, (iv) knowledge distillation (Lu et al., 2022; Smith et al., 2021; Zhang et al., 2020; Lee et al., 2019b; Hou et al., 2018; Li & Hoiem, 2017) which involves using distillation on the previous model, and (v) model rectification (Zhou et al., 2022; Liu et al., 2021; Yu et al., 2020; Belouadah & Popescu, 2019; Castro et al., 2018) which involves training a model on new classes. Our work is inspired by brain-inspired replay (BI-R) (Van de Ven et al., 2020) which is a generative experience replay method, where we assumes that a task model $D$ is trained to solve the classification problem and a generator model $G$ is trained to learn generating pseudo-data points for the past learned tasks to implement generative replay. BI-R uses additional regularization terms for improved performance:

- Replay through feedback, where the generator $G$ is merged into the task model $D$.

- Conditional replay: the latent distribution is modeled as a Gaussian mixture model and allows conditional sampling based on class label $Y$ to build a balanced dataset.

- Gating: a different subset of neurons are disabled in the backward process of the generator.

- Internal replay: the first several layers are pretrained and frozen. Replay parameters are updated only on the trainable layers.

Among all the above methods, BI-R without internal replay is the best-performing model under the strict setting where limited resources is available. Our method is inspired by this model and it provides additional model regularization using dynamic objective weighting through feedback.

# 3 PROBLEM DEFINITION: CLASS INCREMENTAL LEARNING AND GENERATIVE REPLAY

## 3.1 CLASS-INCREMENTAL LEARNING

Class-incremental learning is a continual learning scenario where a model is sequentially trained on new classes without access to previously seen class data. At each task timestamp $t$, only a previously unseen subset of data $X$ and class labels $Y$ are made available to the model $D_t$, defined as $X_t$ and $Y_t$. The objective of this learning scenario is to minimize the aggregate task loss term at time $T$ is:

$$L_{\text{CI}} = \sum_{t=1}^{T} L_{\text{task}}(X_t, Y_t; D_T) \tag{1}$$

The task commonly used in continual learning is classification because emergence of new classes is very natural. For a classification task at timestamp $t$ provided by inputs $X_t$ and their respective labels $Y_t$, and given a classifier $f$ parameterized by learnable weights $\theta$, the task loss is defined as:

$$L_{\text{task}}(X_t, Y_t; D_t) = -\sum_{i=1}^{N} Y_{t,i} \log(f(X_{t,i}; D_t)) \tag{2}$$

We study a **strict setting** in our work. The scalability of incremental learning algorithms is heavily impacted by growing space complexity. Existing state-of-the-art (SOTA) models utilize growing model architecture, or growing storage for exemplars, which makes the performance comparison less meaningful since the brute force space-scaling can overshadow the actual contribution of continual learning mechanisms. However, the human brain maintains a relatively stable amount of neural connections throughout life, with constant development and pruning of neural connections. Understanding what mechanisms can improve continual learning performance and making a fair comparison between different strategies thus has to be conducted under a strict setting: (i) constant model size, (ii) no pre-training dataset, and (iii) no memory buffer storage for past task data. Such a requirement aims to simulate the setting where humans can learn new tasks without forgetting old ones even without exact memory storage, externalization for memorizing data points, or an increasing number of brain connections when more classes are learned. For comparative results, we exclusively selected algorithms that fall into this category, in addition to the traditional unrestricted settings. We think this setting resembles the constraints that under which the human brain works.

## 3.2 GENERATIVE REPLAY

**Generative replay** is a strategy where a generative model $G$, such as a Variational Autoencoder (VAE), is trained to approximate the input data distribution of previously encountered tasks. During the training phase of a new task, instead of relying solely on the original data, the model also rehearses using pseudo-data generated by the generative model. This approach mitigates the issue of catastrophic forgetting of earlier tasks which is especially severe in class-incremental learning because of inter-class diversity. The overall objective in generative replay consists of two main components balanced by hyperparameter $\alpha$: task loss $L_{\text{task}}$ and replay loss $L_{\text{replay}}$:

$$L_{\text{GR}}(X_t, Y_t; D_t) = L_{\text{task}}(\hat{X}_{1:t-1}, \hat{Y}_{1:t-1}; D_t) + L_{\text{task}}(X_t, Y_t; D_t)$$
$$+ \alpha\{L_{\text{replay}}(\hat{X}_{1:t-1}, \hat{Y}_{1:t-1}; G_t) + L_{\text{replay}}(X_t, Y_t; G_t)\}, \tag{3}$$

Where $\hat{X}_{1:t-1}$ denotes the generated samples from $G_{t-1}(\hat{Z}_{1:t-1})$, where $\hat{Z}_{1:t-1} \sim \mathcal{N}(0, I)$. $\hat{Y}_{1:t-1}$ is predicted labels by $D_{t-1}(\hat{X}_{1:t-1})$. The replay Loss focuses on both accurate data reconstruction and maintaining a regularized latent space for sampling. Variational autoencoder (VAE) is the generative model often used and the objective for training it is denoted as:

$$L_{\text{replay}} = L_{\text{VAE}} = \underbrace{\mathbb{E}_{q_\phi(z|x)}[\log p_\theta(x|z)]}_{\text{Reconstruction Term}} - \underbrace{D_{\text{KL}}(q_\phi(z|x)||p(z))}_{\text{Regularization Term}} \tag{4}$$

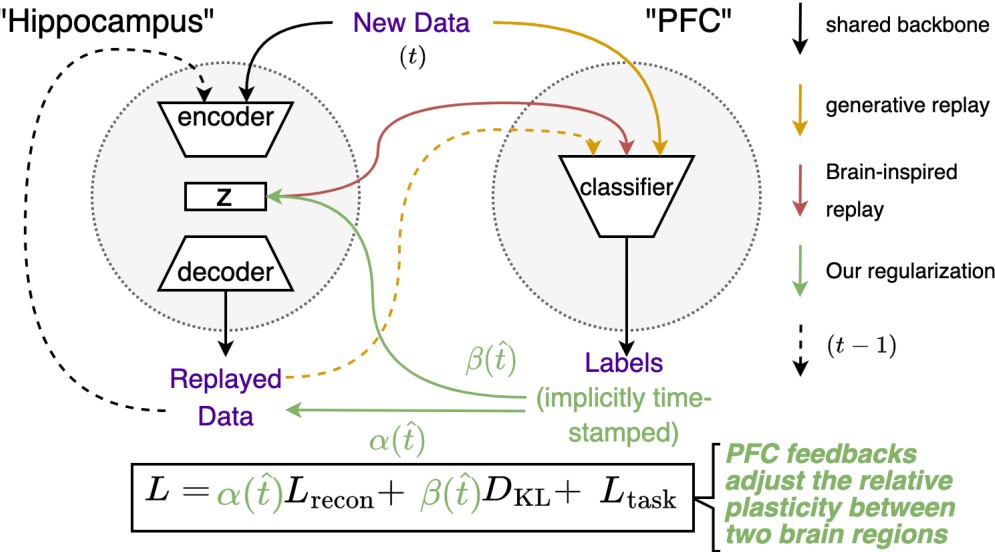

Figure 1: Our regularization method applied to the generative replay pipeline. We introduce additional feedback to adjust the relative plasticity between the Hippocampus and the PFC. Two regularization parameters can also adjust the emphasis on episodic memory or schematic memory.

Where $x \in X$ is the input data. $\theta$ and $\phi$ are the parameters of the VAE's encoder and decoder, the combined is $G$. $q_\phi(\mathbf{z}|\mathbf{x})$ denotes the encoder's approximation of the posterior. $p_\theta(\mathbf{x}|\mathbf{z})$ is the likelihood of reconstructing the data given the latent representation. $p(\mathbf{z})$ is the prior distribution of the latent space, often assumed to be a standard Gaussian distribution.

## 4 PROPOSED METHOD

A high-level description of our method is to adjust the loss function through feedback. We add additional decaying hyperparameters to the generative replay loss function and use the time information inferred from the classifier to guide such decay. We present our detailed modifications and methods.

### 4.1 $\beta$-VAE GENERATOR

Instead of using a plain VAE as the replay generator, we use a $\beta$-VAE. $\beta$-VAE is a VAE variant which allows the trade-off between the KL term and the reconstruction term:

$$L_{\beta\text{-VAE}} = \underbrace{\mathbb{E}_{q_\phi(z|x)}[\log p_\theta(x|z)]}_{\text{Reconstruction Term}} - \underbrace{\beta D_{\text{KL}}(q_\phi(z|x)||p(z))}_{\text{Regularization Term}} \tag{5}$$

### 4.2 DYNAMIC CONTINUAL LEARNING LOSS, WITH TIME-DEPENDENT $\alpha, \beta$

To control the relative strength of the $\beta$-VAE's loss and the task loss in our continual learning setting, we modified the hyperparameter $\alpha$ introduced in 3.2 to only control the reconstruction part of the replay loss, and combine it with $\beta$. In addition, we make these hyperparameters dependent on the inferred time-stamp $\hat{t}$ which is the elapsed time since the first time seeing some label $y$, inferred from the output $\hat{y}$ of the task model $D$. The total loss function $L$ can be formulated as:

$$L = \alpha(\hat{t})L_{\text{recon}} + \beta(\hat{t})D_{\text{KL}} + L_{\text{task}} \tag{6}$$

The block-diagram of our method within the generative replay framework is presented in Figure 1.

### 4.3 INFER TIME-STAMP $t$

When the numerical class label increases through the arrival of new classes, the ordinal time stamp is implicitly encoded in the label. For example, consider a binary digit classification setting with 5 tasks, each consisting of two digits. When we compare class 0 to class 9 when learning the last task, the model implicitly can predict that 0 has been seen a long time ago rather than recently. This implicit time prediction can be enhanced by the decaying sampling quality among classes after recursive sampling and replaying. We infer the time $\hat{t}$ as:

$$\hat{t}(\hat{y}) = \# \text{ tasks seen so far} - \lceil \frac{\hat{y}}{\# \text{ classes per task}} \rceil \tag{7}$$

Where $\hat{y}$ is the predicted numerical class label. Since this label is predicted by the classifier, we bridge the feedback pathway by using this information to control the generative path's loss function.

### 4.4 SCHEDULING $\alpha$ AND $\beta$

The best-performing schedule is an exponentially decaying function with a small lower bound (0.2 through exhaustive search). The process of finding the schedule and how it is related to neural science is discussed in the ablative experiments and discussion section. The general functions are:

$$\alpha(\hat{t}) = (1-a) \cdot e^{-k_\alpha \hat{t}} + a \quad \text{and} \quad \beta(\hat{t}) = (1-b) \cdot e^{-k_\beta \hat{t}} + b$$

Our empirical exploration indicates that our best performing $k_\alpha$ is around 1 and $k_\beta$ is around 10, with $a = b = 0.2$. We also show in the discussion section that such parameter choices correlate well with the qualitative function of the brain, demonstrating our method benefits from similar mechanisms.

## 5 EXPERIMENTS

### 5.1 EXPERIMENTAL SETUP

The comparative and ablative experiment settings are identical to the ones used by BI-R (Van de Ven et al., 2020) for fair comparison against exiting works. We used MNIST, permMNIST, and CIFAR100 datasets for experiments. Note that more complex benchmarks are not used for generative replay because the size of VAE should become very large to generate high-quality samples. The generative backbone is a convolutional variational autoencoder (CVAE). For BI-R, a classification head is attached to the latent space $z$ of the generative model. For baselines, "joint" is when all tasks are made available to the model which serves as an upperbound, and "none" is the setting where no continual learning method is applied to the classifier which serves as a lowerbound, and the generator is not used in this case. We report the average accuracy across all classes at the end of training. In addition, we also visualize the replayed samples and their modified FID score. The modified FID score is used in BI-R to evaluate the quality of replay samples. Smaller FID indicates better quality.

### 5.2 COMPARATIVE RESULTS

We tested different schedule settings, with brain-inspired replay backbone on several class-incremental learning settings: MNIST(5 tasks), permutedMNIST (10 tasks), and CIFAR-100 (10 tasks). Table 1 is the average accuracy benchmark at the end of training for several class-incremental learning strategies. Unrestricted settings allow model growth, Our method outperformed the best-performing method in strict settings. We conclude that our method is effective.

We then evaluate the qualities of the replayed samples generated by BI-R and our method shown in Figure 2. We measure the modified FID score of the generated samples concerning the original dataset. The lower the score, the higher the quality. Although generated samples do not have to be high-quality to increase the overall task performance, it is a good measurement of memory retention capability. We can see that our method leads to generating more diverse samples.

Figure 3 shows the accuracy curve throughout training. In the left subfigure, our method (solid lines) gradually outperforms the baseline BI-R(dotted lines) as more tasks are incoming. Our method shows improved accuracy on all replayed tasks by trading off the accuracy on the current task. As

| Strategies | Method | MNIST(5-TASK) | permMNIST(10-Task) | CIFAR-100(10-Task) |
|---|---|---|---|---|
| **Unrestricted Settings** | | | | |
| Baselines(ResNet18) | Joint | - | - | 80.4 |
| | None | - | - | 8.00(±0.18) |
| Growing Network | DyTox | - | - | 77.15 |
| | DER | - | - | 77.18 |
| **Strict Settings** | | | | |
| Baselines(CNN) | Joint | 98.20(±0.02) | 98.06(±0.04) | 53.96(±0.20) |
| | None | 19.98(±0.02) | 17.79(±0.7) | 8.00(±0.18) |
| Regularization | EWC | 19.92(±0.08) | 27.55(±0.15) | 8.49(±0.05) |
| | online EWC | 19.96(±0.26) | 33.2(±0.11) | - |
| | SI | 20.08(±0.32) | 24.33(±0.21) | 8.51(±0.01) |
| Replay | LwF | 20.08(±0.12) | 21.00(±1.19) | 9.83(±0.13) |
| | GR | 82.59(±0.37) | 91.53(±0.06) | 6.22(±0.12) |
| | BI-R | 91.50(±0.06) | 97.15(±0.03) | 21.01(±0.24) |
| | BI-R + Our method | **94.63(±0.04)** | **97.98(±0.03)** | **24.16(±0.30)** |

Table 1: Average accuracy for different class-incremental learning Strategies

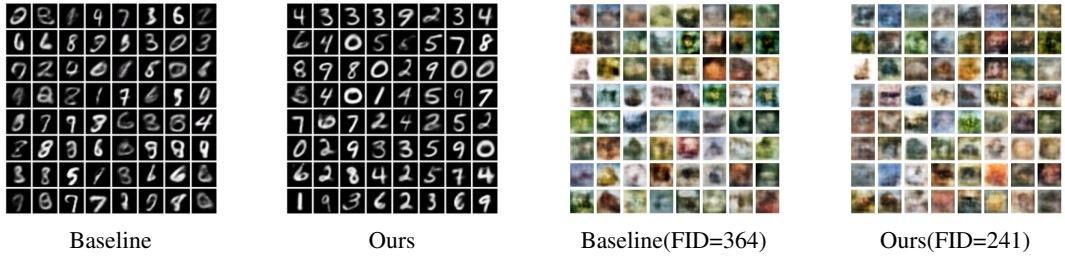

| Baseline | Ours | Baseline(FID=364) | Ours(FID=241) |

Figure 2: Replayed samples at the end of training. From left to right: MNIST (BI-R baseline and ours), CIFAR100 (BI-R baseline and ours).

a result, the effect of such a trade-off will be dominated by the increasing number of tasks, and the average performance margin will expand as shown in the right figure. This tradeoff is discussed further in the discussion section.

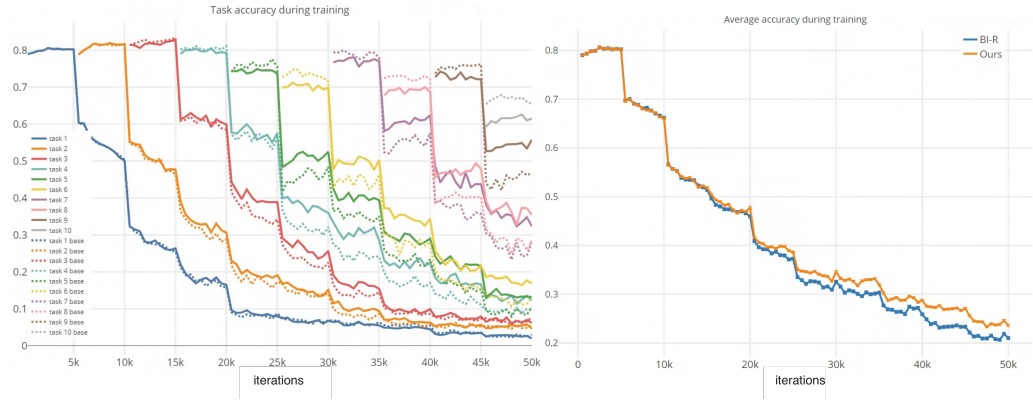

Figure 3: The comparison between BI-R and our method. We compare the accuracy of different tasks (left) and the average accuracy (right) throughout training. Our method has significantly less forgetting on previously learned tasks, by trading off the performance on the current task.

# 6    ABLATIVE STUDY

We conducted several control experiments when BI-R is used as the baseline model:

(i) Only adjust $\alpha$: **(1)** time-independent, **(2)** time-aware

(ii) Only adjust $\beta$: **(3)** time-independent, **(4)** time-aware

(iii) Adjust both $\alpha$ and $\beta$: **(5)** time-independent, **(6)** time-aware

(iv) Adjust both $\alpha$ and $\beta$ according to our decay schedule: **(7)** inferred time from true class labels, **(8)** inferred time from predicted class labels

Figure 4 visualizes the ablative study results. Subfigures (2)(4)(6)(8) denote are our method. In (2)(4)(6), the labels are values we set for replayed tasks parameter $\alpha$ and $\beta$. In (8), the labels are the lowerbound (if $< 1$) or upper bound (if $= 2$) for our decaying schedule function for ($\alpha$ and $\beta$). We observe that (8) is the best setting groups, and the green curve is the best individual setting in this group. (1) and (3) exclusively study the effect of a different constant of $\alpha$ and $\beta$. (5) studies the combined effect of different $\alpha$ and $\beta$. (7) studies the effect of decaying loss function in general through time. We show the relative improvements in the final average accuracy of them in Figure 4. (1) and (2) show that only changing $\alpha$ will not improve the performance. (3) shows a smaller $\beta$ in general will improve the performance. (4) shows a small $\beta$ for replayed classes and 1 for new tasks can further improve the performance on top of (3). (5) shows changing both $\alpha$ and $\beta$ at the same time will not improve the performance. (6) shows a small $\alpha$ and $\beta$ for replayed classes and 1 for new tasks can further improve the performance compared to (4). (8) shows following a decay schedule dependent on the predicted label performs better than a decay schedule dependent on the true labels (conditional labels in conditional replay).

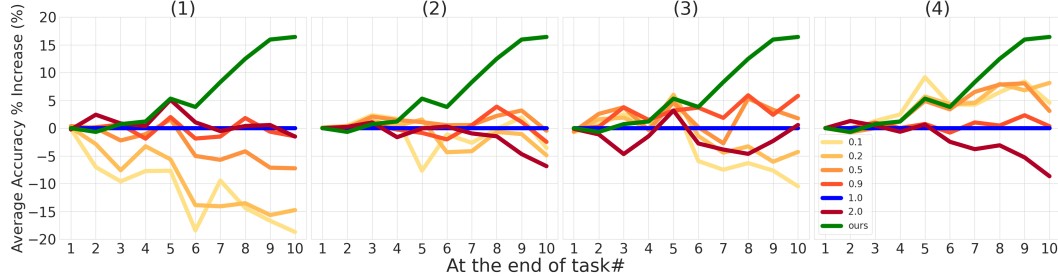

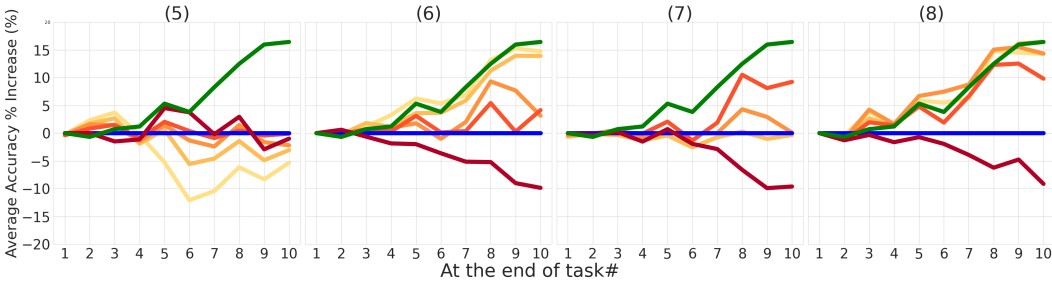

Figure 4: Ablative experiments (1) to (8) on CIFAR-100 (10-TASK). Blue denotes BI-R, (8) is the proposed method, and green is the best setting of our proposed method. The label is the hyperparameter we use for the value of $\alpha$ and $\beta$ in (1) to (6), and the lower or upperbound in (7) to (8).

Our method also shows that the accuracy improvement depends on the length of the task horizon. The more tasks incoming, the better improvements we can expect. This trend is crucial for real-life applications since the task horizon will be infinite.

A parallel analysis of the FID score of the replayed samples is shown in Figure 5. We observe that our method is showing improved sample quality across all groups.

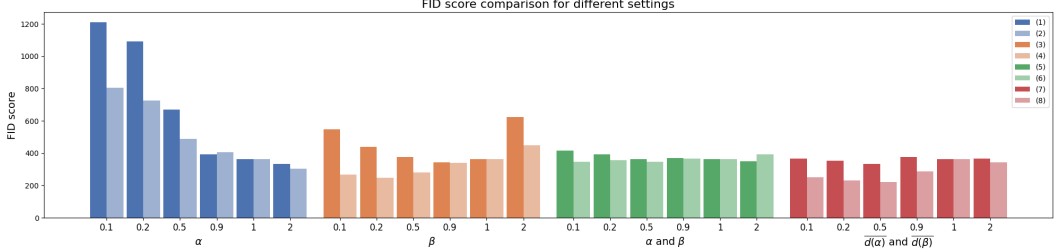

Figure 5: Ablative experiments (1) to (8) on CIFAR-100(10-TASK) compararing replayed samples quality. Light colors indicate our method improves sampling quality in general, the best setting is (8), and the best decay lower-bound in setting (8) is 0.5.

## 7 ANALYTIC EXPERIMENTS

We visualized the latent space of our model compared with BI-R in Figure 6 using the MNIST benchmark. We see that data points are more clustered in each class's cluster, especially the early classes for BI-R. This loss of variance due to the discrepancy in the regularization time horizon under the same regularization amplitude results in low-variance early classes and high-variance new classes. Our method shows a more balanced variance across classes arriving at different times.

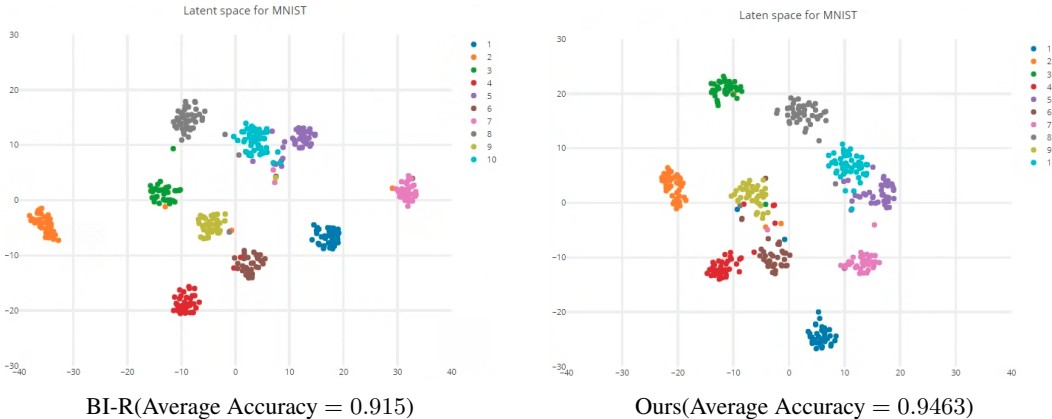

BI-R(Average Accuracy = 0.915)      Ours(Average Accuracy = 0.9463)

Figure 6: The latent space visualization on MNIST dataset. Our method shows more spread-out clusters with higher variance within each class.

## 8 DISCUSSION

### 8.1 ROLES OF $\alpha(\hat{t}), \beta(\hat{t})$ AND THEIR NEUROSCIENTIFIC INTERPRETATION

$\alpha(\hat{t})$ modulates the strength of the reconstruction loss and $\beta(\hat{t})$ modulates the strength of KL-divergence regularization. The combined changes the relative learning rate between the generator and the classifier. The relative balance between these components has neuroscientific significance when likened to the interplay between episodic and schematic memories in the brain and neural plasticity under novel and familiar stimuli:

- **Generation Loss and Hippocampus Synaptic Plasticity**: A higher emphasis on the combined generation loss simulates the increased synaptic plasticity of the hippocampus. Exposure to novel stimuli can lead to such an increase, manifesting as changes in synaptic strength, which is foundational for encoding new memories. When exposed to familiar stimuli, the hippocampus is less likely to experience extensive synaptic changes.
- **Reconstruction Loss and Episodic Memory**: A higher $\alpha$ in loss mirrors the hippocampus's emphasis on forming episodic memory, retaining detailed specifics of experiences.

- **KL Divergence and Schematic Memory**: A higher $\beta$ regularizes the latent space to align with a prior, resembling the hippocampus's emphasis on forming schematic memories that abstract and generalize knowledge across experiences.

A qualitative relative strength of these mechanisms in the brain can be shown in Table 2. The plasticity strength corresponds to our decaying function for both $\alpha$ and $\beta$. Episodic memory formation corresponds to the decaying function of $\alpha$ with $k_\alpha \approx 10$. Schematic memory formation corresponds to the decaying function of $\beta$ with $k_\alpha \approx 1$

| Experience Type | Plasticity Strength | Episodic Memory Formation | Schematic Memory Formation |
|---|---|---|---|
| Novel Experience | High | High (initial encoding of unique experiences) | Moderate (building new schemas or adjusting existing ones) |
| Recent Memory Replayed | Moderate (reconsolidation) | Moderate (reinforcing or updating recent episodic memories) | Moderate (reinforcing or updating existing schemas) |
| Distant Memory Replayed | Low (most consolidation has occurred) | Low (older episodic memories are more cortex-dependent) | Low to Moderate (schemas are largely established and may not need the hippocampus as much) |

Table 2: Qualitative Strength of Hippocampal Processes for Different Experiences. Our hyperparameter schedules qualitatively follow these trends.

By scheduling $\alpha$ and $\beta$, our model can adjust 1. the synaptic plasticity of the hippocampus. 2. the emphasis on forming episodic memory or schematic memory. This flexibility is reminiscent of the brain's recognition and consolidation processes, where the hippocampus is more active when seeing novel experiences, stores clear memories, and, over time, the neocortex abstracts generalized knowledge, forming schematic memories. In the context of continual learning, striking the right balance ensures effective learning and retention across tasks.

## 8.2 LIMITATIONS:

our method is only applicable to the generative replay framework which means that when scaling to a more complicated dataset using a small latent dimension is challenging. From our experiments, the largest improvement is around $15\%$, which is still far from the methods under unstrict settings or joint baselines. In addition, the performance is limited by the complexity of our neural network backbone. For hyperparameter searching and comparison purposes, the generative model complexity is far from SOTA models such as latent diffusion models and GANs. Our best schedule parameters are also found empirically and the exponential decay function is a sensible guess.

## 8.3 POTENTIAL IMPROVEMENT:

The schedule we used is not learnable, but such a schedule can be learned using reinforcement learning given the feedback. Since our method is designed as a plug-in method for any unified model with generative and discriminative components, we can apply this method to more settings other than generative replay where such models are used. In addition, the time-related information extraction is related to the time-embedded U-Net backbone used in diffusion models. Better segmentation of this time information might be possible with such a segmentation neural network backbone.

## 9 CONCLUSION

We presented an incremental learning method rooted in the mechanisms of neural plasticity and memory encoding, specifically adjusting the parameters $\alpha(\hat{t})$ and $\beta(\hat{t})$ through feedback within the joint classification and generative replay framework. The empirical evaluations reinforced the effectiveness of this strategy, especially when benchmarked under a strict setting. Notably, the enhanced quality of replayed samples and a more balanced distribution in the latent space across different class arrival times stand as testaments to the method's efficacy. The insights gathered from human memory systems and their incorporation into artificial neural models are invaluable. This cross-disciplinary approach has led to enhanced performance in the continual learning domain. By bridging neuroscience and artificial intelligence, we pushed the existing brain-inspired methods and provided more insights into the parallelism of human intelligence and machine intelligence.

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

## A   APPENDIX

Our code will be made available on GitHub.

