# OpenReview forum: "CLASS-INCREMENTAL LEARNING USING GENERATIVE EXPERIENCE REPLAY BASED ON TIME-AWARE REGULARIZATION"
_ICLR.cc/2024/Conference — Submitted to ICLR 2024_

### Official Review · Reviewer_ur9z · 2023-10-29

**Soundness:** 3 good
**Presentation:** 3 good
**Contribution:** 2 fair
**Rating:** 6
**Confidence:** 4

**Summary:**

The authors propose an extension to continual-learning methods based on generative-experience-replay, called 'time-aware regularization'. The method involves using time-dependent regularization coefficients for both the beta parameter in beta-VAE and a trade-off coefficient between the generator's losses on current task samples vs replay-generated samples. The time variable is inferred from the predicted class id, since classes are learnt incrementally (and thus the predicted class ids reflect the order in which they were learnt).

The authors focus on class-incremental learning, and present an experimental validation on three benchmarks, pairwise MNIST, permuted-MNIST, and CIFAR-100 (10 tasks).

**Strengths:**

The authors tackle a challenging problem (class-incremental continual learning) in a strict setting, constraining the growth of the models and not allowing the storage of any subset of previous samples.

The paper is well written and easy to follow.

**Weaknesses:**

The authors should describe the experimental tasks in more detail, since the benchmarks mentioned can be implemented in many different ways. For example:
    - Are output units added incrementally for new classes, or are the same original output units mapped to multiple classes?
    - Permuted-MNIST is typically used for task-incremental continual learning. The authors mention 10 tasks, but it is not clear whether that refers to 10 random permutations (each with its 10 digits), or to a single permutation (with the tasks being the individual digits). If the benchmark is used in a task-incremental way (10 permutations each with the full 10 digits available for the same permutation), then the EWC result with an accuracy of just 27% is puzzling, since the original paper -in that setting- achieves over 97-98% accuracy.
    - Same for CIFAR-100; it is not clear what the 10 tasks refer to. Is incremental classification performed on 10 tasks at a time? The authors should be clear about it.

Minor:
    - some typos, e.g., "The idea is to stor and the", or "laten space" in Fig. 6(b), or section 8.2 missing capital letter at the beginning of the sentence "our method is only applicable".

**Questions:**

The 'brain-inspired replay' arrow in fig. 1 is not clear? It would be helpful to have a more detailed caption.

---

### Official Review · Reviewer_EsQv · 2023-10-29

**Soundness:** 3 good
**Presentation:** 3 good
**Contribution:** 2 fair
**Rating:** 3
**Confidence:** 2

**Summary:**

The authors target the modal collapse challenge in generative replay methods for class incremental learning. By adjusting the weights of $L_{reconstruct}$ and $D_{KL}$, they aim to dynamically regularize against losing the diversity of early classes (i.e., modal collapse). Experimental results support the authors' claim that the proposed method improves memory retention while increasing the average performance over continually arriving tasks.

**Strengths:**

1. The addressed problem is crucial.
2. The methodology is straightforward and clearly presented.
3. Empirical results are promising.
4. Despite numerous hyper-parameters, case (8) in Figure 4 suggests low sensitivity to $a$ and $b$.

**Weaknesses:**

1. The rationale for the scheduling in Section 4.4 is unclear. Although the feedback concept aligns with models like ACGAN, the authors' specific scheduling approach lacks theoretical or intuitive foundations. Displaying $\alpha(\widehat{y})$, $\beta(\widehat{y})$, and their ratio, coupled with a discussion, might offer better insight.
2. Figure 3 reveals the proposed method's diminishing plasticity over time, particularly in tasks 9 \& 10. The unaddressed plasticity-stability dilemma limits its real-world applicability. Also, the introduction should cover the plasticity stability dilemma.
3. The FID score is redundant; better stability metrics like BWT exist~\cite{diaz2018don}.
4. Section 6 can benefit from presentation polishing.
5. Section 7 is not an analytical analysis.
6. The extended neuroscience connection can be condensed to make room for implementation details, enhancing reproducibility.
7. The study can also benefit from comparisons with more recent SOTA methods as BI-R is 2020 and EWC is even older.

**Questions:**

1. In Section 6, does "time-aware" refer to the Section 4.4 schedule?
 2. What does the red curve in Figure 4 represent, what is the upper bound, and why is its performance low?
 3. How were $K_{\alpha}$ and $k_{\beta}$ chosen?
 4. In Figure 4, the y-axis indicates a relative value. Is it relative to BI-R?
 5. What do $\overline{d(\alpha)}$ and $\overline{d(\beta)}$ in Figure 5 represent?
 6. How does the proposed method contribute to the more fundamental issue of generative models modal collapse during recursive ~\cite{shumailov2023model}.

References
[1] Natalia D ́ıaz-Rodr ́ıguez, Vincenzo Lomonaco, David Filliat, and Davide Maltoni. Don’t forget, there is more than forgetting: new metrics for continual learning. arXiv preprint arXiv:1810.13166, 2018.
[2] Ilia Shumailov, Zakhar Shumaylov, Yiren Zhao, Yarin Gal, Nicolas Papernot, and Ross Anderson. Model dementia: Generated data makes models forget. arXiv preprint arXiv:2305.17493, 2023

---

### Official Review · Reviewer_yfoE · 2023-10-30

**Soundness:** 1 poor
**Presentation:** 2 fair
**Contribution:** 2 fair
**Rating:** 3
**Confidence:** 4

**Summary:**

In this paper, the authors propose a new method that works in a constrained class incremental scenario. The constraints are: constant model size, no pre-training dataset and no memory buffer for storing past tasks data. Given these constraints, and based on the brain-inspired replay method proposed in [Van de Ven et al.], the author presents a time-aware regularization to adjust the reconstruction and distillation loss dynamically. The authors run experiments on multiple benchmarks, showing promising results against other comparative methods. They also add suitable ablation and discussion sections.

**Strengths:**

- Proposing methods inspired by how the brain works is an excellent approach to finding alternatives to commonly used methods. Exploring new ideas, different from the common ones is appreciated.
- The section on problem definition is helpful to understand the scenario and learn/understand the background of the method.
- The summary presented in Section 8 is handy. However, it could be helpful if those comments are complemented with experiments showing that it is actually happening to what is expected.

**Weaknesses:**

- My biggest concern about this paper is the scenario and the restrictions the authors are using. On one hand, they motivate the idea of not using a memory buffer due to privacy concerns. However, for a generative model to work in this scenario, it is a requirement that it generates data as similar as possible to past distribution, creating an issue about the data replicability. On the other hand, they motivate the idea of using a generative model to have a fixed amount of memory allocated, something that is normally done and easily achieved with memory-based methods.
    - I understand the first motivation if the generative model would work in a latent space. With a generated vector should be unlikely to replicate sensitive information.
- Assuming that generative-based methods can be used in some scenarios where memory-based methods can not. The limitations on the dataset that can be represented with generative methods limit them to particular problems.
    - I recommend finding an application, example or scenario where generating the input can help to present a more stimulating idea.
- Figure 1 needs to be clarified. I don't know what each color of the arrows represents.
- The example in section 4.3 needs to be clarified. Class 0 was seen in the first task?
    - Can we say that "t" is like the task ID?
    - What do you mean by "predicted numeral class label"?
- I appreciate works that do ablation studies. However, it isn't easy to understand what the colors and figures represent.

**Questions:**

- It is unclear how the idea of the bi-directional interplay between the PFC and the hippocampus is related to the approach proposed in the paper. I understand that the alpha and beta are conditioned by something similar to a task id, instead of being conditioned by each other. Is this idea correct, or is there something that I miss?
    - In other words, it needs to be clarified how the ideas mentioned in the second paragraph of the introduction are reflected in the proposal.
- In your opinion, does the proposed method limit the plasticity of the model as more classes are seen?
- Have you explored the idea of generating latent representations instead of inputs? There are works in this direction that meet the requirements to be used in the strict setting used in your experiments.
- Have you considered the idea of using a conditional VAE? To condition given the class or even the task Id?
    - Can this provide the mode with the ability to generate a proper representation of all previous classes with only one distribution?
- Why is LwF considered a replay-based method?

---

### Official Review · Reviewer_2WvB · 2023-10-30

**Soundness:** 3 good
**Presentation:** 3 good
**Contribution:** 1 poor
**Rating:** 3
**Confidence:** 3

**Summary:**

The paper focuses on class-incremental learning and introduces a time-aware regularization method aimed at adjusting the importance weights of the generative loss within the objectives.  The proposed method fine-tunes the weights associated with the reconstruction loss and regularization term in the final objective. These weights are depended on the infer time-stamp (order) of tasks. Empirically, earlier tasks tend to have smaller weights in both the reconstruction and regularization losses. Combining with Brian-inspired Relay (BI-R), the proposed method demonstrates competitive performance in class-incremental learning across several datasets.

**Strengths:**

The paper studies a practical and important problem: class-incremental (continual) learning, focusing on improving the generative replay methods.

The proposed method, which employs a time-aware regularization to tune the weights of generative loss, provides a reasonable method to improve the performance of generative replay.

Overall, the paper is well-structured and effectively communicates the details of the proposed method.

Furthermore, in the experimental evaluation, the proposed method demonstrates superior performance compared to previous approaches, as reported in the paper.

**Weaknesses:**

1) The paper's novelty appears limited, as similar ideas have been explored in prior works. For instance, BioSLAM [1] employs a time-decay factor to reduce the importance of older samples  (see section 5.B.3 in [1]), resembling the time-aware regularization introduced in this paper. Could you clarify the distinctions and unique contributions of this paper in comparison to the time-decay importance mechanism in [1]?

2) The structure of the time-aware regularization method presented in section 4.4 seems tricky. It would be helpful to conduct ablation studies on different time-aware regularization functions, e.g., linear decay, polynomial decay, and exponential decay (as demonstrated in the paper).

3)  The scope of the experiments is somewhat limited. It might be beneficial to expand the experimental setup to include different numbers of tasks, such as a 5-task or 20-task configuration on CIFAR-100 datasets, as discussed in [2].

4) For datasets with high complexity and resolution, such as ImageNet, the quality of sample generation using VAE might not meet acceptable standards, which could potentially restrict the performance of the proposed method. It would be helpful to conduct experiments on large datasets like ImageNet, as discussed in [3].

[1] P. Yin, et al. "BioSLAM: A Bioinspired Lifelong Memory System for General Place Recognition." IEEE Transactions on Robotics (2023).
[2] J. James, et al. "Always be dreaming: A new approach for data-free class-incremental learning." ICCV. 2021.
[3] X. Hu, et al. "Distilling causal effect of data in class-incremental learning." CVPR, 2021.

**Questions:**

1) What is the difference and uniqueness of the paper compared to the time-decay importance in BioSLAM [1]?

2) What is the reason behind selecting the exponential decay term for time-aware regularization in section 4.4? How does this form compare to a linear-decay approach?

3) It might be beneficial to expand the experimental setup to include different numbers of tasks, such as a 5-task or 20-task configuration on CIFAR-100 datasets.

4) It would be helpful to conduct experiments on large datasets like ImageNet.

---

### Meta-Review · Area_Chair_Dcqn · 2023-12-04

**Metareview:**

The manuscript presents an improvement to the generative replay based class incremental learning, through a time-aware regularization method.

Strength:
1. The paper is clearly articulated (Reviewer 2WvB, yfoE, EsQv and ur9z)
2. The method is straightforward and has some novelty (Reviewer 2WvB, yfoE, EsQv and ur9z)
3. Evaluation shows that the performance of the method is good. (Reviewer 2WvB, yfoE, EsQv and ur9z)

Weakness:
1. The paper's novelty appears limited, as similar ideas have been explored in prior works. The distinctions and unique contributions of this paper in comparison to the time-decay importance mechanism in prior studies (Eg: BiioSLAM) has to be discussed. ((Reviewer 2WvB)
2. Experiments are not sufficient

**Justification For Why Not Higher Score:**

Weakness:
1. The paper's novelty appears limited, as similar ideas have been explored in prior works. The distinctions and unique contributions of this paper in comparison to the time-decay importance mechanism in prior studies (Eg: BiioSLAM) has to be discussed. ((Reviewer 2WvB)
2. Experiments are not sufficient
3. Reviewer's concerns have not been addressed.

**Justification For Why Not Lower Score:**

N/A

---

### Decision · Program_Chairs · 2024-01-16

Reject